# Decoding Urban Industrial Complexity: Enhancing Knowledge-Driven Insights via IndustryScopeGPT

## ABSTRACT

Industrial parks are critical to urban economic growth. Yet, their development often encounters challenges stemming from imbalances between industrial requirements and urban services, underscoring the need for strategic planning and operations. This paper introduces IndustryScopeKG, a pioneering large-scale multi-modal, multi-level industrial park knowledge graph, which integrates diverse urban data including street views, corporate, socio-economic, and geospatial information, capturing the complex relationships and semantics within industrial parks. Alongside this, we present the IndustryScopeGPT framework, which leverages Large Language Models (LLMs) with Monte Carlo Tree Search to enhance tool-augmented reasoning and decision-making in **I**ndustrial **P**ark **P**lanning and **O**peration (IPPO). Our work significantly improves site recommendation and functional planning, demonstrating the potential of combining LLMs with structured datasets to advance industrial park management. This approach sets a new benchmark for intelligent IPPO research and lays a robust foundation for advancing urban industrial development.

## CCS CONCEPTS

• **Applied computing** → *Computer-aided design*; • **Human-centered computing** → *Collaborative and social computing*.

## KEYWORDS

Urban Knowledge Graph, Industrial Park Planning and Operation, Large Language Model Agent

## 1 INTRODUCTION

Industrial parks are key engines driving urban economic growth and centers of innovation within cities. They connect the economy, living environments, and environmental sustainability, fostering the integration of technological innovation and urban life [29]. However, many face a significant imbalance between industrial growth and urban service provision, leading to unsustainable development patterns [2]. This imbalance highlights the urgent need for strategic and scientific planning and operation of industrial parks. Such operation requires a comprehensive consideration of local economic levels, infrastructure, and industrial foundations, aiming to provide optimization suggestion, public service facility site recommendation, and comprehensive industrial zone planning [3].

Traditional approaches, often based on empiricism and outdated surveys, fail to dynamically integrate rich urban data for deep analytical insights. [1, 21].

With the progress of information collection technologies, multi-source and multi-modal urban data is rapidly accumulating. The advancement of artificial intelligence further enhances intelligent urban services and tasks such as traffic management [33],urban planning [22], urban function prediction [37], public safety [15], and site recommendation [10]. However, the application of intelligent operation and planning in industrial parks still holds significant untapped potential. The emergence of Large Language Models (LLMs) [19] heralds a significant shift, as they possess powerful language reasoning and in context learning capabilities [27]. This adaptability makes them exceptionally suitable for tackling the complexities of urban systems, thus paving the way for new approaches to unified and adaptive solutions in intelligent industrial park planning and operation (IPPO).

**Challenge 1: How to effectively construct an industrial park dataset capturing complex relationships and semantics?**
Current datasets often overlook the detailed needs of industrial parks, focusing mainly on geographical features and neglecting multi-modal data like cultural and socio-economic elements. Industrial park data is typically multi-source and heterogeneous, including information sources such as enterprises, government websites, statistical yearbooks, street views, etc. Although these data are not explicitly linked, they inherently share attributes and spatial relationships that constitute rich semantic information. To thoroughly evaluate an industrial park's development, it is essential to leverage multidimensional information. Knowledge graphs, by organizing and integrating these diverse data sources, offer a streamlined framework that enhances data management and application in industrial park settings.

**Challenges 2: How to adapt LLMs for industrial park KG?**
Industrial park knowledge graphs differ from general knowledge graphs by being heterogeneous, incorporating image, textual, numerical, and geospatial data with intricate entity relationships. Unlike LLMs focused on textual tasks, they require real-time geospatial data transmission and computations, necessitating graph databases and tools for live interactions (Figure 1). To address the limitations of static knowledge in LLMs, integrating real-time knowledge graph databases and leveraging retrieval-augmented generation (RAG) methods [7] offer a promising solution. Further research is needed to enhance LLMs' reasoning by bridging the gap between knowledge bases and user queries effectively.

**Challenge 3: How can LLMs flexibly and interpretably excel in diverse IPPO tasks?**
Intelligent planning and operation of industrial parks require the unification of numerous tasks. Traditional urban models, often trained on specific datasets for particular tasks, lack the necessary flexibility for broader applications. For example, the features

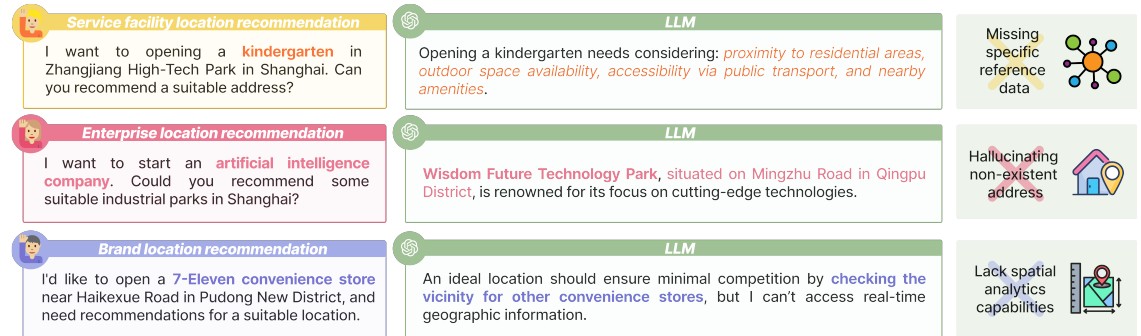

**Figure 1: The challenges in Integrating LLMs for IPPO Solutions.**

considered for situating financial institutions versus restaurants differ significantly, making it costly to retrain models. Additionally, popular methods like CNN-based spatial representational learning [16] can diminish the interpretability and credibility of decisions. Improving decision explainability through clear metrics and continuous reasoning is essential for effective industrial park planning. While recent advancements in LLM multi-step reasoning [26] and autonomous agents [36] show promise for flexibility and interpretability, these often operate in isolation and could benefit from integration with external graph databases to enhance their reasoning capabilities.

To tackle the challenges mentioned, this paper introduces a pioneering work that constructs a multi-modal, multi-level large-scale industrial park knowledge graph, IndustryScopeKG. By extracting entities from diverse multi-modal data sources and combining domain knowledge, a substantial industrial park knowledge base with various spatial and semantic relationships has been built. The IndustryScopeGPT framework is introduced to enable LLMs to dynamically adapt to the structure of the knowledge graph and enhance decision-making capabilities through Monte Carlo Tree Search and reward information. The performance of the framework in IPPO tasks is validated through the development of IndustryScopeQA benchmark, demonstrating the reliability and advantages of the framework in handling domain tasks. Our contributions are summarized as follows:

- We innovatively release the first open-source, multi-modal, multi-level (spatial and semantic level) large-scale knowledge graph dataset, IndustryScopeKG, for diverse tasks in industrial parks.
- We introduce the IndustryScopeGPT framework, which enhances LLMs' planning, action, and reasoning capabilities through the integration of external graph databases and various tools, including Monte Carlo Tree Search for optimal reasoning paths. This framework represents the inaugural implementation of LLMs' fusion with spatial computing and dynamic reasoning on graph databases containing external geographic data.
- We introduce the IndustryScopeQA benchmark to evaluate the IndustryScopeGPT framework's performance. Experiments on site selection and industrial park planning confirm

that the IndustryScopeKG dataset and framework enhance the efficiency and adaptability of LLMs.

## 2 RELATED WORK

### 2.1 Urban Intelligence and Dataset

Researchers utilize deep learning models to extract representations from urban data like satellite images, Points of Interest (POI), grids, and road networks [23, 30, 31]. However, the lack of interpretability in these models hampers understanding and restricts their practical application in urban settings. Additionally, these models tend to focus on specific tasks, lacking generality and generalization abilities. Urban knowledge graphs, organizing urban entities into a complex graph, have become crucial in modern smart cities [17]. Challenges include limited datasets tailored for specific tasks and the absence of publicly available urban knowledge graphs, hindering research progress. Initiatives like OpenSiteRec [10] and UUKG [18] employ heterogeneous graphs to enhance brand site recommendations and urban spatio-temporal predictions. ReCo [6] offers datasets on residential community layouts with precise vector coordinates, benefiting architecture and urban planning. The reliance on single data sources in existing datasets limits the integration of multi-modal urban data, impacting the understanding of urban diversity and the analysis of complex urban systems.

### 2.2 LLMs Reasoning

LLMs exhibit strong capabilities through enhanced reasoning abilities via logical structures like chains [26], trees [35], and graphs [20]. Recent advancements enable large models to access internal and external knowledge for improved decision-making [7]. LLMs are increasingly used as central controllers for autonomous agents with human-like decision-making skills [25]. They have spurred innovation in urban research, such as developing mobility strategies [24], simulating disease spread [28], urban planning [39] and complex spatio-temporal question answering [11]. However, these methods primarily focus on textual spatio-temporal features, neglecting multi-modal urban knowledge.

**Table 1: Urban Dataset Comparison**

| Dataset | Image (Street View) | Socioeconomic Indicators | Geographical Data | Semantic Feature | Multi-scale | Size | Open Source |
|---|---|---|---|---|---|---|---|
| WANT [14] | | ✓ | ✓ | | | 100* | |
| $O^2$-SiteRec [32] | | | ✓ | | | 39,465 | |
| UrbanKG [12] | ✓ | ✓ | ✓ | | | 17,407,159 | |
| UrbanKGent [17] | | ✓ | ✓ | | | 67,978 | ✓ |
| OpensiteRec [10] | | ✓ | ✓ | | | 6,170,925* | ✓ |
| KnowSite [13] | | ✓ | ✓ | ✓ | | 920,504 | ✓ |
| UUKG [18] | | | ✓ | | ✓ | 1,490,680 | ✓ |
| **IndustryScopeKG(Ours)** | ✓ | ✓ | ✓ | ✓ | ✓ | 51,684,939 | ✓ |

\* denotes the approximate number w.r.t.the corresponding paper
*For taubular data,'size'counts the number of data points, and for KG, it counts the number of triples in total.*

## 3 INDUSTRYSCOPEKG

### 3.1 Data Collection and Pre-processing

*3.1.1 Data Acquisition.* We opted to acquire multi-source spatio-temporal data from the Shanghai, China, considering the richness and availability of information relevant to industrial parks Figure 2. The data sources mainly come from three aspects:

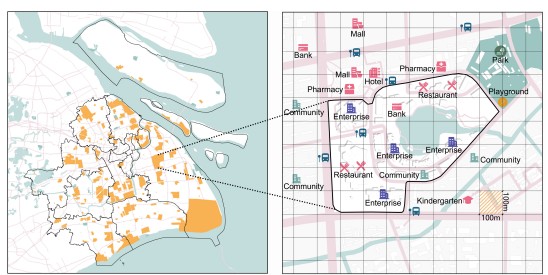

**Figure 2: Park Vectorization and Grid Processing.**

**Urban Geospatial Data:** Includes (1) detailed building footprints, areas of interest (AOI), and mobility data based on mobile positioning collected from Baidu Map; (2) points of interest (POI) data and public transport station data from Amap.

**Corporate Data:** Includes (1) industrial and commercial registered enterprise data retrieved from enterprise information inquiry website Qichacha; (2) enterprise patent data and software copyright data from the National Intellectual Property Administration; (3) listed company data, state-owned enterprise data, high-tech enterprise data, small and medium-sized enterprises (SMEs) in technology, and overseas (cooperative) company data from Macrodatas.

**Socioeconomic Data:** Includes (1) census data and regional GDP data from government reports; (2) up-to-date information on housing prices from Beike . Note that all of the data is collected from open-source data sources to fulfill the ethical regulations.

*3.1.2 Data Pre-processing.* Before constructing the IndustryScopeKG, significant data preprocessing was necessary, focusing on geospatial tasks like geocoding and spatial positioning. We manually outlined the vector boundaries of each industrial park using maps of Shanghai's industrial parks, converting text addresses into geocoded latitude and longitude coordinates. Additionally, we standardized the coordinate systems of the multi-source data to the Baidu system for accurate positioning within industrial park boundaries. Integrating visual models, we employed semantic segmentation of street view images to calculate the green view index, used object detection to assess permeability, and trained models on street view charm value based on expert ratings.

### 3.2 Knowledge Graph Construction

*3.2.1 IndustryScopeKG.* **Definition.** We define the graph $G = (E, S, Y)$, where $E = \{e_1, e_2, \ldots, e_{|E|}\}$ is a set of $|E|$ entities, $S$ denotes the set of relational triples, and $Y$ encompasses the set of attributional triples. Specifically,

**Relational Triples:** $S \subseteq E \times R \times E$ represents a collection of triples that delineate the relationships between entities, with $R$ constituting a set of $|R|$ distinct relations. For instance, "Company – Located In – Industrial Park".

**Attributional Triples:** $Y \subseteq E \times A \times V$ constitutes a set of triples indicating the attributes of entities, where $A = \{A_1, A_2, \ldots, A_{|A|}\}$ represents a collection of $|A|$ attributes, with each attribute $A_i \in A$ paired with a corresponding set of values $V_i \in V$. For example, "Industrial Park – Number of Companies – 500" (Figure 4).

*3.2.2 Relational Triples Extraction.* **Entity Extraction.** For the IndustryScopeKG, we extract entities from 8 major categories and 32 sub-categories. The major categories include: **(1) Industrial Parks,** encompassing 264 industrial parks in Shanghai. **(2) Grids,** which are 128,866 fine-grained spatial grids derived from gridding industrial parks. **(3) Grid Dominant Functions,** identified from calculating the dominant Points of Interest (POI) within the grids. The data is then adjusted using AOI information, resulting in 15 types of grid functions such as business offices, commercial services, residential areas, among others. **(4) POI,** serving as the basic functional units and places, including 15 POI sub-categories such as residential, green spaces, business offices, commercial services, etc. **(5) Enterprises,** including entities of 1,058,656 enterprises within the parks. **(6) Enterprise Industries,** comprising primary industries, secondary industries, tertiary industries, and scope of

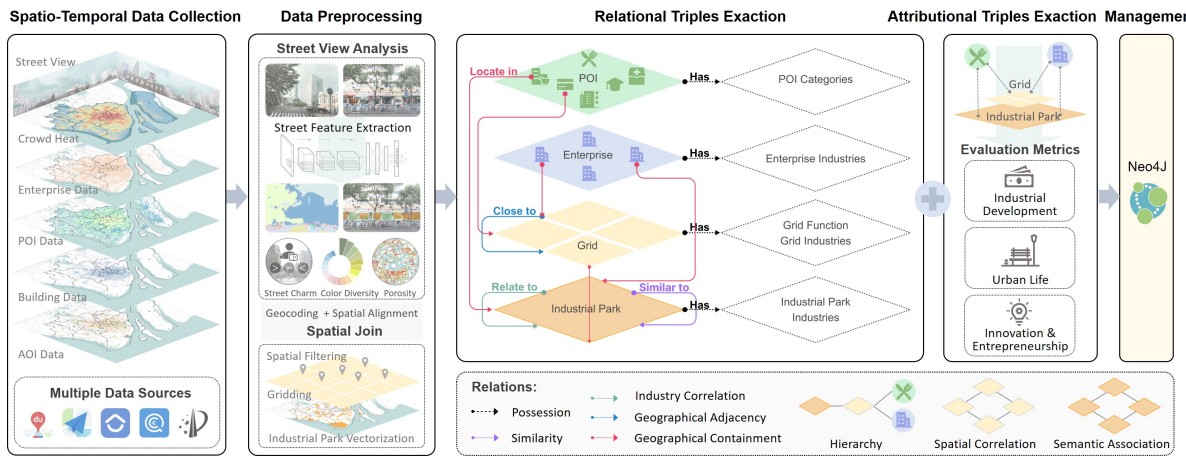

**Figure 3: IndustryScopeKG Construction Pipeline.**

operations, divided into 4 sub-categories. **(7) Industrial Park Industries,** covering planned industries, leading primary industries, leading secondary industries, leading tertiary industries, and leading scope of operations, divided into 5 sub-categories. These are established based on the frequency and significance of their occurrence within the enterprises in the parks. For industrial park $P$, with enterprises set $E_P$, and industry categories $C_{1,P}, C_{2,P}, C_{3,P}$ corresponding to primary, secondary, and tertiary sectors, the leading industry and scope of operation are identified as:

$$L_{k,P} = \operatorname*{argmax}_{c_k \in C_{k,P}} \left( \frac{|\{e \in E_P | e \text{ is categorized as } c_k\}|}{|E_P|} \right) \quad (1)$$

$$O_{s,P} = \operatorname*{argmax}_{s \in S_P} \left( \frac{|\{e \in E_P | \text{scope } s \text{ is listed in } e's \text{ operations}\}|}{|E_P|} \right) \quad (2)$$

where $L_{k,P}$ is the leading industry for the category level $k$ and $O_{s,P}$ represents the predominant scope of operations based on the enterprise's registered activities within $P$.

**(8) Grid Industries,** following a similar classification structure to industrial parks, are divided into four sub-categories.

**Relation Extraction.** We extracted spatial and semantic relationships. Spatial relationships include geographical containment and adjacency; semantic relationships cover the similarity between industrial parks, the correlation of industries within industrial parks, and possession.

**Geographical Containment:** This explains how one entity is located within another entity, categorized into three types: (1) POI/Enterprise Located in Grid, (2) POI/Enterprise Located in Industrial Park, and (3) Grid Located in Industrial Park, facilitating detailed spatial analysis.

**Geographical Adjacency:** This refers to the spatial proximity between entities, detailed in two types: (1) Grid Adjacent to Grid: Identifies proximity and adjacency between grids. (2) Industrial Park Adjacent to Industrial Park: Similarly, this specifies the proximity between parks.

**Similarity:** Indicates the resemblance between industrial parks, expressed as Industrial Park Similar to Industrial Park. We compute

the embeddings for each industrial park's unique features and assess park similarities based on cosine distance.

**Industry Correlation:** Denotes the connection between industrial parks based on industry characteristics, expressed as Industrial Park Related to Industrial Park. We derive embeddings for each park's industry-related aspects, such as planned industries and leading industries. An industry correlation link is forged between two parks if their industry similarity surpasses a threshold of 0.9.

**Possession:** Connects entities with what they possess. For example, Industrial Park Has Planned Industries, Grid Has Leading Primary Industries, and Enterprise Has Scope of Operations.

*3.2.3 Attributional Triples Extraction.* To address the challenge of managing diverse entity attributes in a graph database, we use attributional edges for efficient navigation and analysis. Establishing a comprehensive evaluation system is crucial for the robust development of industrial parks, involving an in-depth exploration of urban vitality and industrial park evaluation frameworks [40]. By analyzing regions like Silicon Valley based on People, Economy, Society, Place, and Governance, detailed data reveals the importance of a quantifiable indicator system covering Industrial Development, Urban Life, and Innovation and Entrepreneurship.

**Industrial Development:** Includes the number of enterprises, large-scale enterprises, average registered capital, state-owned enterprises, listed companies, and industrial agglomeration, match degree of planned industries, working population, GDP, office building area and its proportion, etc.

**Urban Life:** Includes accessibility to residential functions, public services, transportation stations, functional diversity, density of public service functions, of function richness, function compatibility, average housing price, work-life balance index, etc.

**Innovation and Entrepreneurship:** Includes the number of newly registered enterprises, technology-based SMEs, high-tech enterprises, overseas (cooperative) companies, patents and copyrights, and industrial diversity, accessibility of innovation support functions, population with higher education, density of financial services, density of research functions, etc.

**Table 2: The Statistics of Entities**

| Basic Statistics | Industrial Park | Grid | Grid Dominant Function | POI | Enterprise | Total |
|---|---|---|---|---|---|---|
| Count | 264 | 128,866 | 15 | 112,931 | 1,058,656 | 1,300,732 |
| (Leading) Industries | Primary | Secondary | Tertiary | Scope of Operations | Planned | |
| Industrial Park | 202 | 258 | 261 | 261 | 70 | 1,052 |
| Grid | 1,142 | 6,270 | 10,281 | 20,246 | / | 37,939 |
| Enterprise | 18 | 90 | 392 | 891,814 | / | 892,314 |

**Table 3: The Statistics of Triples**

| Relation | Head & Tail Entity | Triple Records |
|---|---|---|
| Locate in | (POI, Grid) (Enterprise, Grid) (POI, Industrial Park) (Enterprise, Industrial Park) (Grid, Industrial Park) | 2,516,160 |
| Adjacent to | (Grid, Grid) (Industrial Park, Industrial Park) | 488,401 |
| Similar to | (Industrial Park, Industrial Park) | 3,765 |
| Related to | (Industrial Park, Industrial Park) | 10,687 |
| Has | E.g., (Industrial Park, Planned Industries) (Grid, Leading Scope of Operations) (Grid, Dominant Functions) | 4,252,341 |
| Attribution | (Industrial Park, Value) (with 111 attributions) (Grid, Value) (with 82 attribution) (POI, Value) (including 15 attributions) (Enterprise, Value) (with 36 attributions) | 44,413,585 |

Within this framework, the industrial park encompasses 74 specific sub-indicators, while the grid is detailed through 48 sub-indicators. By employing a correlative computational strategy, we seamlessly integrate the unique attributes of smaller spatial units, such as individual grids, into the broader analysis of large-scale entities like industrial parks.

*3.2.4 IndustryScopeKG Management.* Following the outlined process, we have constructed a knowledge graph that contains 2,232,037 entities and 51,684,939 triples. To manage this expansive scale effectively, we employ the Neo4j graph database system for storage, querying, and updates. A key advantage of Neo4j is its spatial capabilities, which greatly enhance our ability to perform spatial computations.

# 4 PRELIMINARY

*4.0.1 Problem Formulation.* When addressing user queries, we developed an LLM-driven agent that is capable of generating text responses and interacting with external tools that facilitate interaction with graph databases. Following action-reflection style work [38], we define the agent's action state at each step $t$ as $a_t \in A$, which is a combination of the text generation $\hat{A}_t$ and the tool action $\bar{A}_t \in T$. Such a state pair is represented as $a_t = (\hat{A}_t, \bar{A}_t)$, where thought $\hat{A}_t$ is intended to encapsulate an understanding of key information and guide the subsequent action $\bar{A}_t$. This action is determined by the policy $\pi(\bar{A}_t \mid Q, a_1, o_1, \ldots, a_{t-1}, o_{t-1}, \hat{A}_t)$. The initial input $Q$ includes the user's query, task description, schema, tool instructions, and some few-shot examples. To balance exploration and exploitation in finding the best trajectory, we used Monte Carlo Tree Search (MCTS) [4]. This approach views the large planning space of IPPO-related decision-making tasks as a tree search process. This approach is necessary due to the agent's uncertainty, the improper use of tools or their execution failures, and the potential for better evaluation dimensions or solutions. In this process, each node state is $S = [Q, a_1, \ldots, a_t, o_1, \ldots, o_t]$. The final answer is derived from the output of the last leaf node on the best trajectory. Each iteration of MCTS consists of four steps:

## 4.1 Monte Carlo Tree Search Planner

**Selection:** The process initiates at the root node (initial state), u employing an enhanced UCT (Upper Confidence bounds applied to Trees) [9] algorithm to guide the search towards promising areas for expansion. This approach dynamically balances exploration and exploitation based on aggregated rewards. The core of this refinement is the updated UCT formula:

$$UCT = \bar{X} + (W \times D^n) \times \sqrt{\frac{2\ln(N)}{n}} \quad (3)$$

Where $\bar{X}$ is the average reward of the node, indicating the node's past performance. $W$ represents the initial exploration weight. $D$ is the decay factor, reducing exploration emphasis with each additional visit to encourage more exploitation of the node as it becomes more familiar. $n$ counts the visits to the current node, and $N$ denotes the total visits to the parent node.

The search progresses, picking actions that either resolve the query or demand further exploration until a termination condition is met. The search process terminates either when a solution is found, or upon reaching a maximum depth, preventing overly deep and less relevant explorations. Additionally, in certain tasks, the process can be configured to terminate if a tool is called more than four times consecutively, safeguarding against redundant searches and ensuring every step meaningfully advances towards a solution.

**Expansion:** During expansion, the search widens by generating new child nodes from feedback on executed actions, all recorded in a long-term memory. Each node undergoes a scalar evaluation to aid future node selection, focusing on simulations that highlight the most promising paths. This phase enables the parallel execution of the best N potential actions, thereby expanding the exploration domain and enhancing the decision tree's coverage.

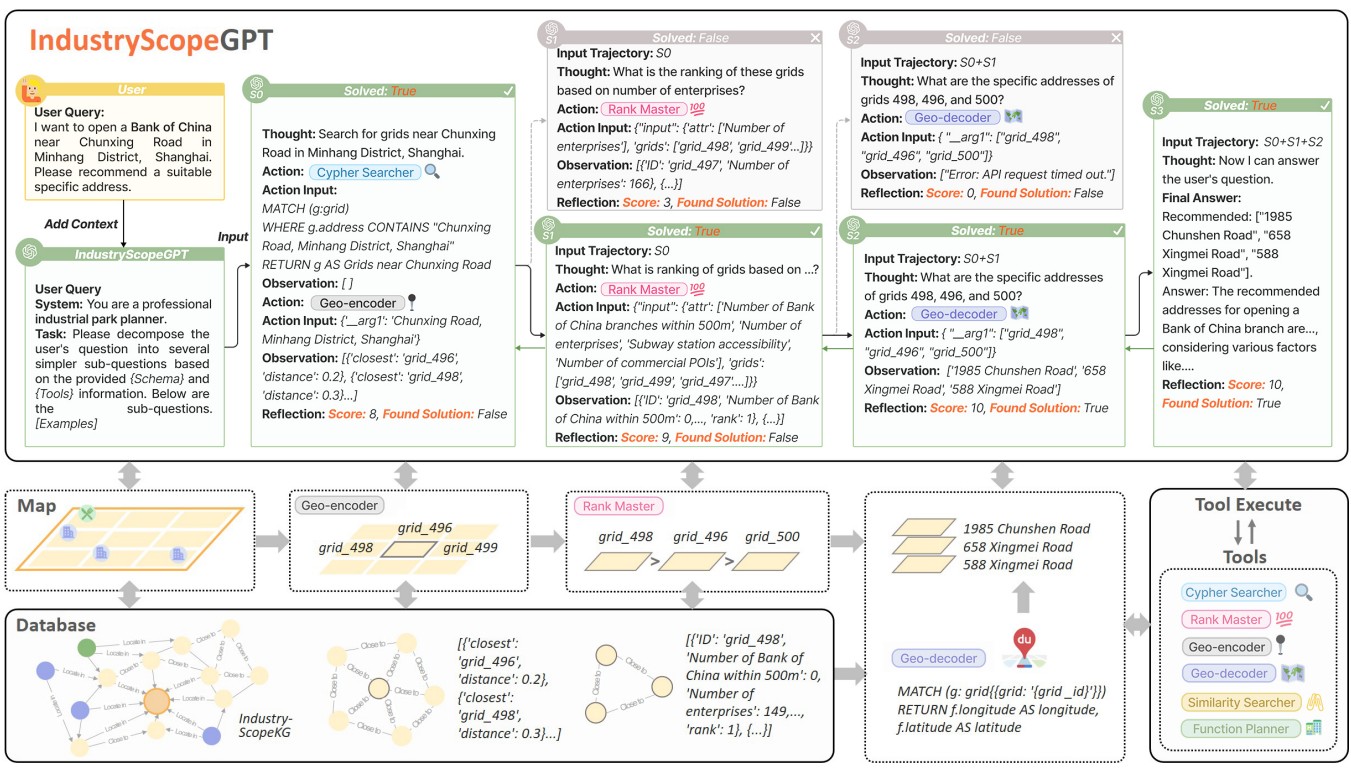

Figure 4: Overview of IndustryScopeGPT.

**Reflection:** Post-action execution, outcomes are assessed, incorporating LLM based self-reflection and external feedback to score decisions. To avoid potential misinterpretations due to a lack of context—limited visibility to previous node information—the reflection process also leverages the trajectory's memory. This includes a history of executed actions and generated outcomes, ensuring a comprehensive contextual understanding.

**Back-propagation:** Following reflection, the values of nodes along the explored paths are updated based on the outcomes of simulations. This critical phase integrates the insights gained from exploration into the decision-making architecture, ensuring that each node incrementally approaches the optimal solution within the complex decision-making landscape of industrial park planning and operation. The back-propagation uses a recursive update mechanism,

$$V(n) = V(n) + \frac{R - V(n)}{N(n)} \tag{4}$$

where $V(n)$ is the current value of node $n$, $R$ is the reward obtained from the simulation, and $N(n)$ is the number of visits to node $n$, incrementally enhancing the tree's accuracy and strategic depth with each iteration.

## 4.2 Decision Support Tools

In crafting a multifaceted decision support system for IPPO, we've developed a suite of nuanced sub-task tools. These tools interact with a Neo4j graph database to facilitate sophisticated queries, analyses, and recommendations, directly addressing the complex needs

of urban planners, investors, and businesses. Here's an overview of the integrated tools and their primary functionalities:

**Cypher Searcher:** Tailored to generate and execute Cypher queries based on user inquiries, this tool sifts through the graph database to provide detailed insights into industrial park attributes, facilities, and demographics, streamlining the data retrieval process for specific user queries.

**Similarity Searcher:** Based on a semantic similarity, this tool searches and recommends parks with high similarity to the user's input, e.g., type of business. By analyzing the match between park features and user's specified business or operational criteria.

**Geo-encoder:** The tool is designed to real-world convert textual addresses into precise geographic locations, specifically identifying relevant grid IDs within a Neo4j graph database. This conversion allows for a seamless mapping of user-provided locations to the spatial framework of the database.

**Geo-decoder:** Conversely, the tool translates geographic grid IDs, obtained from spatial queries or system recommendations, back into human-readable textual addresses.

**Rank Master:** By ranking industrial parks or specific grids based on an array of metrics like accessibility, POI density, and demographic indicators, this tool guides users towards making informed decisions, prioritizing locations that best match their specified criteria.

**Function Planner:** An assistant designed to propose functional planning suggestions for specified areas. Given a grid, it not only retrieves information for the designated and adjacent grids but

**Table 4: Park- Level Site Recommendation**

| Methods and Knowledge | Accuracy | Precision | Recall | F1 |
|---|---|---|---|---|
| GPT-4 w Table/ SE | / | / | / | / |
| GPT-4 w Cypher Searcher | 0.04 | 0.242 | 0.220 | 0.224 |
| CoT w Tools | 0.127 | 0.319 | 0.319 | 0.320 |
| ReAct w Tools | 0.088 | 0.276 | 0.276 | 0.276 |
| **IndustryScopeGPT** w Tools | **0.204** | **0.443** | **0.440** | **0.441** |

**Table 5: Conditional Park-Level Site Recommendation**

| Methods and Knowledge | Accuracy | Precision | Recall | F1 |
|---|---|---|---|---|
| GPT-4 w Table/ SE | / | / | / | / |
| GPT-4 w Cypher Searcher | 0.194 | 0.555 | 0.429 | 0.464 |
| CoT w Tools | 0.166 | 0.488 | 0.488 | 0.488 |
| ReAct w Tools | 0.205 | 0.539 | 0.462 | 0.485 |
| **IndustryScopeGPT** w Tools | **0.233** | **0.659** | **0.566** | **0.590** |

**Table 6: Grid-Level Site Recommendation**

| Methods and Knowledge | Accuracy | Precision | Recall | F1 |
|---|---|---|---|---|
| GPT-4 w Table/ SE | / | / | / | / |
| GPT-4 w Cypher Searcher | 0.064 | 0.368 | 0.368 | 0.368 |
| CoT w Tools | 0.236 | 0.488 | 0.480 | 0.483 |
| ReAct w Tools | 0.232 | 0.509 | 0.428 | 0.457 |
| **IndustryScopeGPT** w Tools | **0.292** | **0.584** | **0.572** | **0.577** |

**Table 7: Conditional Grid-Level Site Recommendation**

| Methods and Knowledge | Accuracy | Precision | Recall | F1 |
|---|---|---|---|---|
| GPT-4 w Table/ SE | / | / | / | / |
| GPT-4 w Cypher Searcher | 0.161 | 0.475 | 0.437 | 0.450 |
| CoT w Tools | 0.127 | **0.550** | 0.395 | 0.433 |
| ReAct w Tools | 0.06 | 0.258 | 0.181 | 0.204 |
| **IndustryScopeGPT** w Tools | **0.194** | 0.492 | **0.483** | **0.487** |

*/ represents mostly zero or near-zero metrics.*

also synthesizes these insights to offer strategic planning advice (Figure 5).

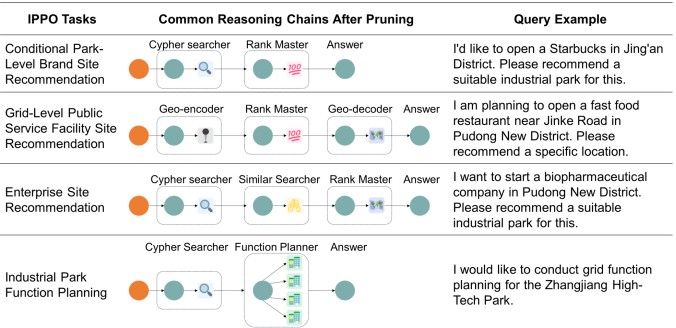

**Figure 5: Typical Reasoning Chain using Tools.**

## 5 EXPERIMENT

To validate the effectiveness of IndustryScopeKG and the IndustryScopeGPT framework, we aim to address the following research questions:

**RQ1:** Can the data and graph structure provided by IndustryScopeKG effectively impact the performance of LLMs on IPPO?

**RQ2:** How does IndustryScopeGPT perform on tasks based on IndustryScopeKG compared to existing LLM prompting paradigms?

**RQ3:** How can IndustryScopeGPT serve industrial park functional planning?

To explore these questions, we designed two typical IPPO tasks.

### 5.1 Multi-spatial Scale Facility Siting Recommendation (RQ1, RQ2)

To improve facility siting flexibility, we introduce a multi-spatial scale recommendation task that leverages LLM's reasoning abilities to analyze and filter spatial attribute features for optimal facility siting, regardless of scale or facility type.

**Dataset.** We created 20,000 siting questions across various spatial scales and facility categories, incorporating attribute data from industrial parks and grids in the IndustryScopeKG. LLM identified 5-8 key evaluation attributes for each question, resulting in three sets. Domain experts selected one set through consensus-building, and top areas were determined using an optimal ranking method, forming question-answer pairs. We used 200 question-answer pairs covering diverse spatial scopes and facility types for test.

**Baseline Methods (RQ1).** To assess the effectiveness of IndustryScopeKG, we evaluated the performance of GPT-4 in various configurations: alone, with tabular input for structured data processing, integrated with a search engine to enhance data retrieval capabilities, and combined with Cypher for querying graph databases. This approach allowed us to compare the impact of different data integration methods on the model's ability to handle IPPO tasks.

**Baseline Methods (RQ2).** For tool invocation, we contrasted IndustryScopeGPT with methods highlighting LLMs' external tool and graph database interaction capabilities. We compared classic prompting with CoT and ReAct methodologies, examining performance across varied siting tasks and spatial scales.

**Experiment Settings.** The same version of gpt-4-0125-preview was used across the experiments, with a graph searcher temperature set to 0 and the QA model temperature set to 0.7.

**Metrics.** We assessed whether the predicted locations matched those in the dataset, measuring Accuracy, Precision, Recall and F1 score.

**Results.** The evaluation demonstrated that GPT-4, whether operating independently or enhanced with tabulated data (where IndustryScopeKG data was structured into tables and embedded into a vector library for retrieval-augmented generation ) or search engine functionalities, performed poorly in complex site recommendation tasks (RQ1). This highlights the significance of our dataset

and graph structure. In contrast, IndustryScopeGPT exhibited superior performance (RQ2), significantly outperforming methods such as those using a Cypher searcher, CoT, and ReAct across most metrics. For example, in the conditional park-level site recommendation, IndustryScopeGPT demonstrated its capability to effectively leverage structured industrial data, achieving a precision of 0.659, and an F1 score of 0.590. This showcases its superior performance in optimizing decision-making for site recommendation tasks.

## 5.2 Industrial Park Functional Planning (RQ3)

This task focuses on the strategic layout optimization of various grid functions within an industrial park, aiming to fulfill the foundational needs and balanced layout of functionalities. The planning process involves a detailed examination of each grid within a specific industrial park, extracting information from targeted and adjacent grids and integrating these attribute insights to assign a function to each grid in industrial park.

**Baseline Methods (RQ3).** Traditional models like LightGBM [8] and GCN [34] were used for comparison to evaluate IndustryScopeGPT's performance in functional planning.

**Dataset.** For training LightGBM and GCN, datasets were prepared matching the input formats required by these models. All 128,866 grid attributes were used as features. For GCN, grid adjacency relationships were used to define edges, and grid functions pre-calculated in IndustryScopeKG, corresponding to 15 categories, served as labels. The data was split into 7:3 for training and testing.

**Metrics.** Given the absence of uniform standards for planning evaluation, we focused on improvements to the existing functional layout. We utilized the Hill numbers formula [5] to quantify functional diversity, which is defined as:

$$H_q = \left( \sum_{i=1}^{S} p_i^q \right)^{\frac{1}{1-q}} \tag{5}$$

where $p_i$ represents the proportion of the $i$-th function type within the park, and $q$ determines the emphasis on abundance. Specifically, $q = 0$ quantifies the Richness, or the total count of distinct functional types, providing an initial diversity level. $q = 1$ corresponds to Shannon Entropy, which accounts for the proportion of each function, offering a more nuanced understanding of diversity. $q = 2$ is tied to Simpson's Index, focusing on the prevalence of dominant functions. These measures are vital for assessing how well the park's space is utilized and ensuring that diverse functions are integrated to support industrial and ancillary activities.

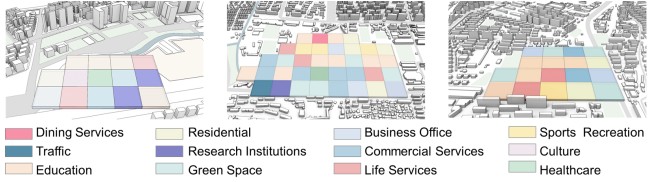

Dining Services · Residential · Business Office · Sports Recreation
Traffic · Research Institutions · Commercial Services · Culture
Education · Green Space · Life Services · Healthcare

**Figure 6: Utilizing IndustryScopeGPT for Grid Function Planning in Parks. (From left to right: Park A, Park B, Park C)**

**Case study.** We focused on three distinct industrial parks: Zhangjiang Artificial Intelligence Island (Park A), Dachang Urban Industrial

**Table 8: Detailed Analysis of Grid Functions by Methods**

| Methods | Grid Functions | q=0 | q=1 | q=2 |
|---|---|---|---|---|
| Real Situation | **Park A**: Green Space | 1 | 1 | 1 |
| | **Park B**: Residential: Sports Recreation: Green Space = 8:3:1 | 3 | 2.39 | 2.11 |
| | **Park C**: Sports Recreation: Green Space: Residential = 12:5:1 | 2 | 1.81 | 1.67 |
| LightGBM | **Park A**: Residential: Green Space = 1:6 | 2 | 1.51 | 1.32 |
| | **Park B**: Residential: Sports Recreation:Green Space = 8:3:1 | 3 | 2.28 | 1.95 |
| | **Park C**: Sports Recreation: Green Space: Residential = 12:5:1 | 2 | 1.81 | 1.67 |
| GCN | **Park A**: Residential: Green Space = 3:4 | 2 | 1.98 | 1.96 |
| | **Park B**: Residential: Green Space = 1:3 | 2 | 1.76 | 1.60 |
| | **Park C**: Residential: Green Space = 4:5 | 2 | 1.99 | 1.98 |
| IndustryScopeGPT | **Park A**: Healthcare: Commercial Services: Business Office: Residential: Education: Culture:Life Services: Research Institutions = 1:1:2:1:4:1:2:2 | 8 | 7 | 6.13 |
| | **Park B**: Education: Commercial Services: Green Space: Residential: Business Office: Sports Recreation: Life Services: Dining Services: Healthcare: Research Institutions: Traffic = 8:6:5:4:3:3:3:1:1:1:1 | 11 | 8.76 | 7.54 |
| | **Park C**: Education: Green Space: Business Office: Residential: Commercial Services: Sports Recreation: Life Services = 6:4:2:2:2:1:1 | 7 | 6.22 | 5.59 |

Park (Park B), and Xinyefang Global Sci-Tech Innovation District (Park C). Each park was chosen for its unique characteristics and the specific challenges it presents in terms of spatial planning. IndustryScopeGPT was tested against two classic models, as well as the real-world scenarios. The results showed that IndustryScopeGPT significantly outperformed these models in the metrics across the three parks. For example, in Park A, known for its focus on green space, IndustryScopeGPT was able to ensure a better balance between green areas and industrial development. In Park B, which combines residential, sports, and green areas, IndustryScopeGPT demonstrated superior ability to integrate diverse functions into a cohesive plan that supports both living and recreational activities. Similarly, in Park C, IndustryScopeGPT promoted an effective synergy between education and commercial activities.

## 6 CONCLUSION

This study presents a transformative approach to intelligent planning and operation of industrial parks through the integration of LLMs and a large-scale multi-modal, multi-level knowledge graph, IndustryScopeKG. Our dataset is pivotal in capturing the intricate relationships and semantics within industrial parks. Introducing the IndustryScopeGPT framework, which incorporates advanced retrieval and reasoning strategies like Monte Carlo Tree Search, our work sets a new standard in the field, enhancing adaptability and interpretability across a variety of IPPO tasks. The empirical findings substantiate the substantial gains in site recommendation performance and functional planning efficacy, achieved through this novel integration.This initiative not only highlights the critical role of IndustryScopeKG in advancing urban industrial applications but also paves the way for further LLM integration into urban spatial development strategies.

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
