# OpenReview forum: "Decoding Urban Industrial Complexity: Enhancing Knowledge-Driven Insights via IndustryScopeGPT"
_acmmm.org/ACMMM/2024/Conference — MM2024 Poster_

### Official Review · Reviewer_i7ep · 2024-05-01

**Rating:** 5
**Confidence:** 3

**Summary:**

This paper introduces IndustryScopeKG, a pioneering large-scale multimodal and multi-level knowledge graph for industrial parks. Additionally, the authors present the IndustryScopeGPT framework, which leverages large language models (LLMs) and Monte Carlo Tree Search to enhance tool-augmented reasoning and decision-making for Industrial Park Planning and Operation (IPPO).

**Strengths:**

(1) This work is highly relevant and meaningful for Industrial Park Planning and Operation, addressing critical challenges in this domain.
(2) The authors have created the first multimodal and multi-scale dataset, IndustryScopeKG, for industrial parks, integrating diverse urban data sources.
(3) The IndustryScopeGPT framework enhances LLMs' planning, action, and reasoning capabilities by integrating external graph databases and various tools.

**Limitations:**

(1) The paper does not clearly specify whether the LLM used in the IndustryScopeGPT framework is a pre-trained open-source model or a model trained from scratch specifically for this task.
(2) The text in the figures, particularly the relationship labels in the knowledge graph, is too faint and difficult to read, which hinders the understanding of the visual representations.
(3) The paper lacks detailed information about the parameters and implementation details of the IndustryScopeGPT framework, which would benefit reproducibility and further research.

**Suitability:**

2

---

### Official Review · Reviewer_6ypv · 2024-05-16

**Rating:** 5
**Confidence:** 2

**Summary:**

The paper introduces IndustryScopeKG, the first open-source, multi-modal, multi-level large-scale knowledge graph specifically designed for industrial parks.  The IndustryScopeGPT framework is introduced to enable LLMs to dynamically adapt to the structure of the knowledge graph and enhance decision-making capabilities through Monte Carlo Tree Search and reward information. The performance of the framework in IPPO tasks is validated through the development of the IndustryScopeQA benchmark, demonstrating the reliability and advantages of the framework in handling domain tasks.

**Strengths:**

1. This knowledge graph captures complex spatial and semantic relationships, a significant advancement over traditional datasets that often focus on geographical features alone.
2. the IndustryScopeGPT framework enhances LLMs’ planning, action, and reasoning capabilities through the integration of external graph databases and various tools, including Monte Carlo Tree Search for optimal reasoning paths.

**Limitations:**

1. The performance of the framework heavily relies on the quality and comprehensiveness of the data in the knowledge graph. Any inaccuracies or gaps in the data could significantly impact the decision-making process. The paper does not discuss potential data quality issues or how they might be mitigated.
2. While the paper presents impressive results for the specific case of Shanghai， it does not fully address the scalability of the framework to larger or more diverse urban areas.

**Suitability:**

3

---

### Official Review · Reviewer_gGk1 · 2024-05-27

**Rating:** 3
**Confidence:** 2

**Summary:**

In this paper, the authors aim to solve the problem of the imbalance between industrial requirements and urban services. To achieve the goal, the authors successfully leverage Large Language Models (LLMs) in conjunction with Monte Carlo Tree Search to enhance tool-augmented reasoning and decision-making processes in Industrial Park Planning and Operation (IPPO). Additionally, the authors release a multi-modal and multi-level knowledge graph dataset of industrial parks. In the experiment, the authors tested the method on the dataset IndustryScopeKG and found that it consistently outperformed existing LLM prompting models.

**Strengths:**

Novelty:
The authors release a new industrial parks dataset, IndustryScopeKG. The introduction of IndustryScopeKG, a pioneering multi-modal, multi-level large-scale industrial park knowledge graph that integrates various data sources to enhance data management and application in industrial settings.
It is really novel and interesting to combine the Large Language Models and the multi-modal and multi-level knowledge graph especially for the industrial settings. This design comprehensively takes the powerful language reasoning and in context learning capabilities into consideration.
Theoretical Approach:
The theoretical foundation of the paper is robust, utilizing advanced AI techniques such as knowledge graphs, Monte Carlo Tree Search, and LLMs to address complex problems in urban industrial settings.
The IndustryScopeKG dataset systematically integrates diverse data sources including socio-economic, corporate, and geospatial information.

Adequate Evaluation:
The authors evaluated the reliability and advantages of the framework of the method using the IndustryScopeQA benchmark.
Clarity:
The paper is well-structured and clearly written, making complex concepts accessible to readers, which is unfamiliar with this domain.
The use of figures, such as the IndustryScopeKG construction pipeline and examples of the decision-making process, enhances understanding of the application in real-world scenarios.
Practical Applications:
The practical applications of this research are significant. The framework not only improves the efficiency of industrial park planning but also has the potential to influence broader aspects of urban development.

**Limitations:**

IndustryScopeKG's data has regional specificities, and in some areas it may not be possible to obtain such fine-grained detailed urban geospatial data, corporate data and socioeconomic data.
And the authors only used this one dataset when evaluating the effectiveness of the model. This dataset is unique data in a specific area, and the results on one dataset cannot well illustrate the effectiveness and universality of the model.
The integration of LLMs with a large-scale, multi-modal knowledge graph can lead to significant computational demands. The author lacks a comparison of the computational complexity with other baselines.
The technical innovation of the proposed model is limited. The authors only use some existing tools to enhance LLM’s planning and reasoning capabilities.

**Suitability:**

3

---

### Official Review · Reviewer_j7fC · 2024-06-04

**Rating:** 3
**Confidence:** 2

**Summary:**

The paper introduces a large-scale and multi-modal knowledge graph for industrial parks integrating diverse urban data leveraging Large Language Models with Monte Carlo Tree Search for enhanced decision-making in industrial park planning and operations. The proposed framework establishes a new benchmark for intelligent industrial park management research.

**Strengths:**

1. The paper presents a very interesting topic and the paper is well organized.
2. Innovative Integration: Pioneering integration of LLMs with structured datasets for urban industrial development.
3. Comprehensive Dataset: The first open-source, multi-modal, multi-level knowledge graph dataset for industrial parks.
4. Advanced Framework: IndustryScopeGPT framework enhances LLMs' capabilities with external graph databases and tools.

**Limitations:**

1. Complexity: The intricate nature of combining various data sources and technologies may present a steep learning curve.
2. Scalability: While promising, scaling the framework to accommodate global industrial park data could be challenging.
3. Generalizability: The current focus on Shanghai's industrial parks may limit the immediate applicability to other regions.
4. The paper has many pictures and tables, while no one table is mentioned/explained in the texts in the paper and also the same situation for some of the figures. Would be nice that the authors could give some texts in the papers linking to the tables/figures.

**Suitability:**

3

---

### Meta-Review · Area_Chair_3ojS · 2024-07-02

**Recommendation:** Accept (Poster)
**Confidence:** 4

**Metareview:**

The AC and reviewers appreciated the rebuttal. AC leans towards acceptance.